# Blue Rubber Bleb Nevus Syndrome (BRBNS): A Rare Cause of Refractory Anemia in Children

**DOI:** 10.3390/children10010003

**Published:** 2022-12-20

**Authors:** Liang-Chun Chen, Chun-Yan Yeung, Chen-Wang Chang, Hung-Chang Lee, Wai-Tao Chan, Chuen-Bin Jiang, Szu-Wen Chang

**Affiliations:** 1Department of Pediatrics, Taipei Hospital, Ministry of Health and Welfare, New Taipei City 242, Taiwan; 2Department of Medicine, MacKay Medical College, New Taipei City 252, Taiwan; 3Department of Pediatric Gastroenterology, Hepatology and Nutrition, Hsinchu Municipal MacKay Children’s Hospital, No. 690, Sec. 2, Guangfu Rd., East Dist., Hsinchu City 300044, Taiwan; 4Department of Hepatology and Gastroenterology, MacKay Memorial Hospital, Taipei City 104, Taiwan; 5Department of Pediatric Gastroenterology, Hepatology and Nutrition, MacKay Children’s Hospital, Taipei City 104, Taiwan; 6Department of Pediatric Gastroenterology, Hepatology and Nutrition, Mackay Memorial Tamshui Branch Hospital, New Taipei City 251, Taiwan

**Keywords:** anemia, blue rubber bleb nevus syndrome, gastrointestinal hemorrhage, refractory

## Abstract

Refractory anemia is not uncommon in pediatric patients, and anemia caused by gastrointestinal tract bleeding should always be kept in mind. Aside from infection or intestinal malrotation related bleeding, vascular malformation should also be considered. Blue rubber bleb nevus syndrome (BRBNS) is a rare vascular disorder consisting of multiple venous malformations. Lack of experience in pediatric BRBNS might lead to delayed diagnosis or misdiagnosis. Herein, we report a case of an eleven-year-old boy with recurrent pallor appearance and weakness diagnosed with BRBNS. After a thorough examination, he was treated with endoscopic polypectomy, and further iron supplements and folic acid. He is now under regular follow-up at our outpatient department. No complication is noted for six months. BRBNS is a rare venous malformation syndrome that mostly involves skin and the gastrointestinal tract. Multidisciplinary approach should be arranged for diagnosis and management. Up to date, no consensus for BRBNS treatment has been reached. Management usually depends on clinical symptoms and severity of damage of involved organs. The options of treatment include conservative, medical, endoscopic, and surgical management.

## 1. Introduction

Refractory anemia, a common disease in pediatric patients, may be accompanied by gastrointestinal (GI) tract bleeding without significant hemorrhagic signs. When refractory anemia is detected, additional comprehensive examinations should be arranged. Patients with GI bleeding may be asymptomatic; however, such bleeding can lead to severe life-threating conditions, including hypovolemic shock. Many diseases can cause GI tract bleeding, including infections, intestinal malrotation-related bleeding, and vascular malformations. Blue rubber bleb nevus syndrome (BRBNS) is a rare vascular disorder consisting of multiple venous malformations. The lesions are most commonly found on the skin and in the GI tract. The most common clinical presentation is symptomatic anemia due to GI bleeding, both overt and occult. The morbidity of BRBNS is low [1]. Reported cases of BRBNS in the pediatric population are extremely rare; the corresponding lack of experience with pediatric BRBNS among clinicians can lead to a delayed diagnosis or misdiagnosis. A delayed diagnosis or misdiagnosis may lead to harmful results due to unnecessary examinations or treatment. Here, we present a pediatric BRBNS case with recurrent iron-deficiency anemia without obvious hemorrhagic symptoms.

## 2. Materials and Methods

An 11-year-old boy came to our outpatient department with pallor and weakness 3 years ago. Severe iron-deficiency anemia (IDA) with a serum level of iron < 10 µg/dL and hemoglobin (Hb) level of 5.3 g/dL was diagnosed. Esophagogastroduodenoscopy was performed and revealed esophagitis, gastritis, and duodenitis with a hemorrhagic spot. He underwent one blood transfusion during admission and took an oral iron supplement for 3 months during follow-up at our outpatient department. However, he was lost to follow-up for 9 months. During this time, slow worsening of the patient’s pallor was observed by his family. Intermittent dizziness, tachycardia, and hypotension were also noted during the month before he visited our outpatient department. Due to increased frequencies of dizziness and tachycardia, he was brought to our hospital for help. During an examination, he mentioned that he had nearly passed out after exercise in the morning before he was brought to our hospital. He reported no abdominal pain, a decreased appetite, diarrhea, melena, bloody stools, tarry stools, or body weight loss. While taking the patient’s medical history, his mother report no family history related to anemia. No history of trauma in the past 3 months was also confirmed by his mother. The patient also had no long-term drug usage history or history of Chinese medicine or alcohol intake. His vital signs revealed tachycardia with hypotension. His heart rate reached 118 beats per minute while his blood pressure was 89/52 mmHg at rest. In physical examinations, facial pallor and a bluish nodule on the tongue (Figure 1) were noted. He and his mother could not recall when this bluish nodule had appeared. No other skin lesion was noted.

Laboratory testing revealed severe iron deficiency anemia (Hb: 4.6 g/dL, mean corpuscular volume [MCV]: 53.6 fL, serum iron: <10 μg/dL, and total iron-binding capacity [TIBC]: 407 μg/dL) but no thalassemia or immune deficiency-related abnormalities. Admission with further examination and management was arranged. Fluid resuscitation with medication was started soon after admission. A blood transfusion with packed red blood cells was given as soon as the blood examinations were finished. Comprehensive assessment for recurrent severe anemia was our next plan. After explaining this to the patient and his family, they agreed to further endoscopic study. Esophagogastroduodenoscopy revealed gastritis, duodenitis, and a suspected angiodysplastic lesion. Abdominal computed tomography (CT) and angiography revealed no significant abnormalities. Due to the refractory anemia and suspected angiodysplastic lesion noted at this time by esophagogastroduodenoscopy, a colonoscopy was arranged and several bluish vascular polypoid lesions were found across the terminal ileum and sigmoid colon (Figure 2).

He was discharged with a ferrum and folic acid supplement and scheduled for regular follow-up at our outpatient department. Dizziness with facial pallor recurred soon without melena or tarry stools after 3 months. Severe anemia with a Hb level of 5.5 g/dL was found again. Following a discussion with the patient’s family, a capsule endoscopic examination was arranged. The examination revealed some lymphoid hyperplasia and dark brown lesions in the ileum. As the patient’s anemia and pallor persisted, retrograde single-balloon enteroscopy was arranged and multiple sessile polypoid/vascular lesions (7–10 mm) were noted in the ileum (Figure 3).

Endoscopic removal of the lesion was completed and accompanied by hemoclips and argon plasma coagulation, which were applied for hemostasis. The pathology report revealed a lymphoid polyp, mucosal tag, and focal lymphoid and hyperplastic polyp at different locations from the colon to the terminal ileum. He was soon discharged with a ferrum and folic acid supplement.

## 3. Results

According to the patient’s clinical manifestations and endoscopic findings, BRBNS was ultimately diagnosed. We discussed the possibility of a gene survey for BRBNS with the patient’s family but they refused due to financial concerns. Thus, our patient underwent a polypectomy with argon plasma coagulation via enteroscopy and no complication was observed for 6 months. He was then lost to follow-up due to the COVID-19 pandemic. During the 6-month follow-up period, the patient experienced no dizziness and his anemia did not worsen based on laboratory testing. We also contacted him via phone and no dizziness or bleeding was reported by the patient or his family.

## 4. Discussion

BRBNS is a rare venous malformation syndrome; to date, only ~350 cases with diverse clinical presentations have been reported globally [2], and only 3 pediatric cases have been reported in Taiwan [3,4,5]. The estimated incidence of BRBNS is low at about 1:14,000 births [6], and it is usually a sporadic disorder. Due to its rarity, a delayed diagnosis leading to potentially severe consequences is possible. Here, we presented a case of BRBNS with refractory anemia in an 11-year-old boy.

Blue rubber bleb nevus syndrome is a rare syndrome including venous malformations that mostly arise in the skin and GI tract. The malformations may be seen in various organs and organ systems, including the liver, spleen, heart, eyes, and central nervous system. The clinical symptoms can vary according to the location of the venous malformations. BRBNS can be detected in infancy if venous malformations are noted on the skin, while lesions located in the GI tract are mostly discovered in adolescence or adulthood because the patient experiences discomfort. No definite etiopathology of BRBNS has been identified. It is considered a sporadic disease caused by mutations in the TEK gene [7], although some reports have revealed an autosomal-dominant inheritance pattern [8,9]. Mutations in TEK may also be associated with other forms of venous malformation that are unrelated to BRBNS [7]. Further study is needed to demonstrate whether there is a strong connection between TEK mutations and BRBNS. In our case, no genetic survey was done due to the family’s financial situation. The clinical symptoms of BRBNS vary greatly. Patients may present with asymptomatic nodules on the skin or severe anemia requiring frequent blood transfusions. The presentation depends on the organs involved, which could allow lesions to progress and increase in number and size over time without being noticed. Although venous malformations might be detectable in infancy, clinical symptoms may occur later in life. It takes time and close observation to diagnose BRBNS. Our patient presented with recurrent anemia without any hemorrhagic signs and was ultimately diagnosed with BRBNS after 3 years of comprehensive examinations.

A multidisciplinary approach should be used in the diagnosis of BRBNS. Aside from taking a complete medical history and conducting physical examinations, multiple other examination types are needed. Blood tests should include a complete blood count, clotting function assay, the determination of serum levels of iron, and stool occult blood assay. A thalassemia test might also be needed if microcystic anemia is detected in blood testing. Imaging for BRBNS also plays an important role in diagnosis. Venous malformations may be present not only on the skin but also in the GI tract. Abdominal CT and magnetic resonance imaging have good sensitivity while abdominal sonography has poor sensitivity due differences in the detectable depth between individuals and operator-dependent differences in image quality. Thorough endoscopic examinations, including esophagogastroduodenoscopy, enteroscopy, and colonoscopy, may be needed because lesions can be found at any location within the GI tract. An endoscopic survey provides not only a thorough view of the GI tract for diagnosis but also for judging the effectiveness of therapy. Our case underwent a thorough examination, including abdominal CT and angiography, which revealed no significant abnormalities. However, in later examinations, we found multiple, suspected, sessile polypoidal vascular lesions (7–10 mm) in the ileum by enteroscopy. Although the pathology report revealed no vascular structure, this could be due to depth limitations during polypectomy considering the thickness of the bowel in pediatric patients. Taking the clinical picture and enteroscopic findings into consideration, we diagnosed the patient with BRBNS. It is difficult to make a diagnosis of BRBNS without thoughtful examinations, as in our case. No study has yet compared the sensitivity and specificity of CT, magnetic resonance imaging, angiography, and endoscopy in patients with BRBNS. Further study is needed to determine the correlation between different imaging modalities and BRBNS. Such information could help clinicians make an earlier diagnosis in cases of BRBNS, make cost-effective diagnostic and treatment decisions, and decrease the possibility of unnecessary investigations.

There is currently no consensus on the management of BRBNS. Treatment planning should be individualized to each patient according to the extent of the disease. Presently, there is no curative treatment for BRBNS. Patients who receive pharmacologic agents or endoscopic polypectomy may experience a recurrence of their clinical symptoms. Conservative treatment with iron supplementation is needed for asymptomatic anemic patients. Occasionally, a blood transfusion may be needed if worsening anemia is observed, even if the patient is taking an iron supplement. Multiple pharmacologic agents have been used with varied responses. Corticosteroids, interferon-alpha, intravenous immunoglobulins, vincristine, and somatostatin analogs have been used in different studies to treat patients with BRBNS. Pharmacologic agents are administered with the intention to reduce the incidence of GI bleeding and the frequency of blood transfusions. There are no guidelines or studies that have made recommendations for or comparisons of the above pharmacologic agents. Possible complications or side effects caused by frequent blood transfusions could be avoided with the help of the above pharmacologic agents. Additional studies are needed to create optimized treatment plans.

Taking the high recurrence rate of GI bleeding into consideration, treatment with anti-angiogenesis therapy should be considered. Aside from anti-angiogenetic agents such as steroids, beta blockers, and interferon, a new agent, sirolimus, has shown promising efficacy [10]. Its use to treat BRBNS was first reported by Yuksekkaya et al. [11] in 2012. It is an immunosuppressant that is mostly used for tumors or diseases related to the immune system. Sirolimus exerts anti-angiogenic effects by blocking the mammalian target of rapamycin signaling pathways. It also decreases vascular endothelial growth factor levels, thereby decreasing vascular malformation. Few studies have reported the efficacy of sirolimus in patients with BRBNS, but most of them have indicated promising effects on cutaneous or GI lesions. The variable doses of sirolimus used in these studies may explain why no standard recommendations exist. Side effects related to sirolimus, such as joint pain, skin rashes or bruises, bleeding, and increased cholesterol levels, are also a concern that requires attention. Additional studies are needed to determine the optimal dosage of sirolimus for patients with BRBNS. Due to the need for clinical improvements in, and the general lack of experience with sirolimus therapy for BRBNS, our patient did not receive sirolimus following a discussion with the patient and his family. Iron supplementation was prescribed and no discomfort was mentioned during follow-up.

Other than pharmacologic treatment, endoscopic intervention plays a major role in patients with recurrent GI bleeding. Endoscopic sclerotherapy, band ligation, or laser photocoagulation are commonly used in adult patients with vascular malformations. Although endoscopic examinations are also commonly used in patients during childhood or infancy, polypectomy, sclerotherapy, band ligation, and laser photocoagulation are not often used compared to patients in adolescence or adulthood. Polyps in pediatric patients are not uncommon; however, most of them are benign without lasting sequelae, and they carry no long-term risk of malignancy. In fact, most polyps do not require removal. Polypectomy via endoscopy may not be advisable in pediatric patients compared to adult patients. Still, given the increased accessibility and maturity of the pediatric polypectomy technique, endoscopic interventions for lesion removal have become more common than traditional surgical bowel resection in pediatric patients suffering from bleeding and frequent symptomatic anemia. Endoscopic interventions may have fewer complications than traditional surgical procedures. Endoscopic polypectomy, hemoclips, and electrocautery can be used according to the number, size, and depth of polyps. Post-polypectomy bleeding and intestinal perforation have been noted and should be closely monitored. Endoscopists should bear in mind that the thickness of the bowel varies with age; thus, extra care should be taken during polypectomy or photocoagulation in younger patients. Patients with an intestinal perforation after polypectomy require hospitalization, broad-spectrum antibiotic therapy, and potentially surgical intervention. After a discussion with the patient and his family, our endoscopist used hemoclips and argon plasma coagulation after polypectomy. No complete vascular structure was identified in the pathology report on our patient, but this may have been related to the tissue samples that were studied. There were no post-polypectomy complications, such as bleeding or an intestinal perforation, noted in our patient during follow-up.

Apart from monitoring the impact caused by the disease itself on the physiological status of our patient, we monitored the impact of the disease on the patient’s mental health and the extent of interference with daily activities. Long-term quality of life and development are important issues when discussing treatment plans for children. Patients and their caregivers may experience significant stress when facing rare diseases; dealing with a rare disease has a large impact on the whole family. Thus, more help is needed than simply providing medical advice for the disease itself; emotional health and financial problems may require attention. Taking all of these aspects into consideration, long-term and comprehensive follow-up is essential for pediatric patients with BRBNS.

## 5. Conclusions

We presented a pediatric case of BRBNS with recurrent iron deficiency anemia without obvious hemorrhagic symptoms. Due to its rarity in the pediatric population and our lack of experience in caring for BRBNS patients, BRBNS was diagnosed after thorough imaging and enteroscopic surveys in our case. Clinicians should keep in mind the possibility of BRBNS in children who suffer from recurrent anemia without GI bleeding and who lack a definite diagnosis. BRBNS has different clinical symptoms depending on the organs affected; thus, it might be difficult to discover in its early stages. A multidisciplinary approach should be implemented as soon as suspicion of BRBNS is raised. Aside from physical examinations and blood testing, imaging for BRBNS plays an important role in making a diagnosis. Additional studies of the sensitivity and specificity of different imaging modalities are needed to make better, cost-effective diagnostic and treatment decisions.

There is currently no consensus on the management of BRBNS. Treatment plans should be individualized according to the extent of the disease. Pharmacologic agents, endoscopic polypectomy, or surgery may be selected for treatment. Treatment plans should be developed considering the best interests of the patient. Owing to the relatively high recurrence rate of bleeding in BRBNS patients, long-term follow-up is recommended. It is also important for the patient and family to become knowledgeable about the disease. It not only helps them monitor the disease’s progress, it also enables the patient and his/her family avoid life-threatening situations.

## Figures and Tables

**Figure 1 children-10-00003-f001:**
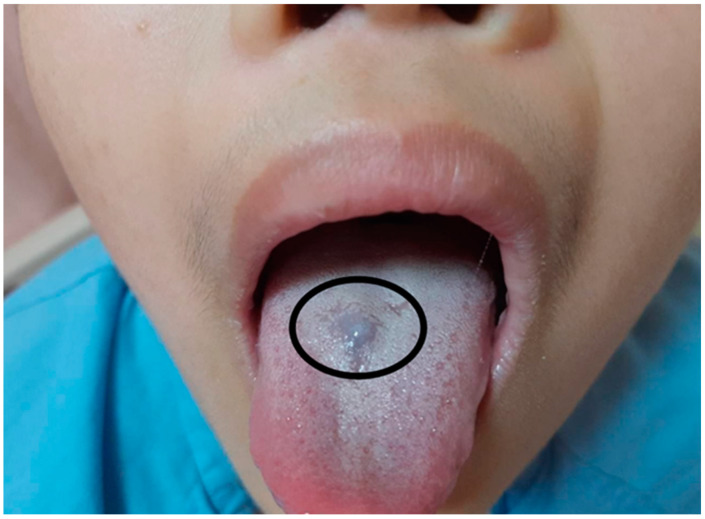
A bluish nodule on the tongue (circled) was seen.

**Figure 2 children-10-00003-f002:**
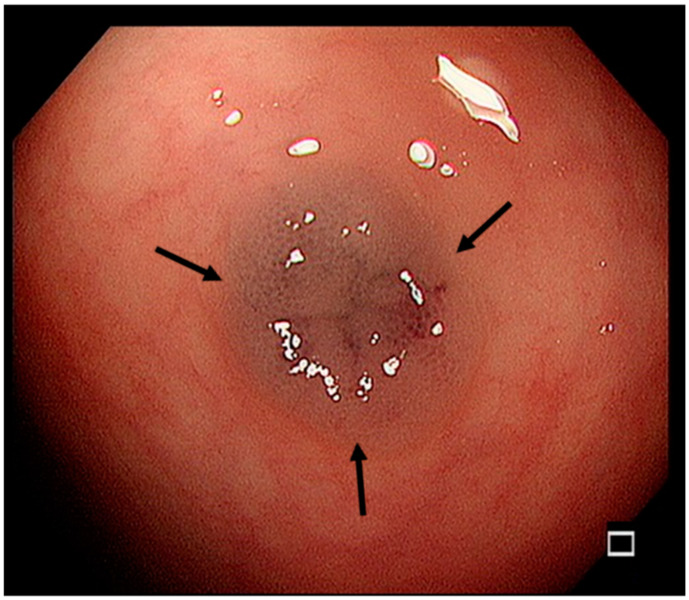
A bluish vascular polypoid lesion found in the terminal ileum (marked with an arrow) as seen by colonoscopy.

**Figure 3 children-10-00003-f003:**
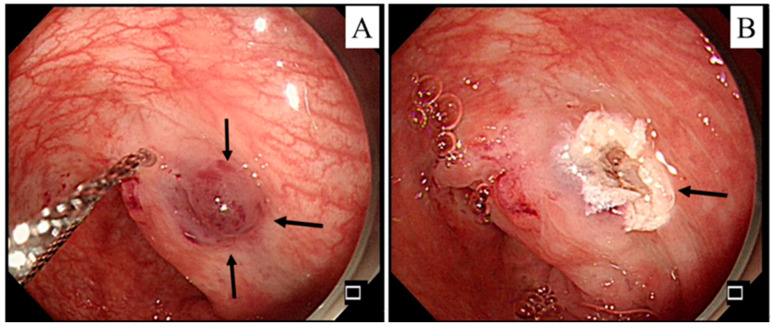
(**A**) A sessile polypoid/vascular lesion was observed in the ileum (marked with an arrow) by retrograde single-balloon enteroscopy. (**B**) The residual lesion after removal by argon plasma coagulation (marked with an arrow).

## Data Availability

Data available on request due to restrictions eg privacy or ethical. The data presented in this study are available on request from the corresponding author. The data are not publicly available due to protection of patient privacy.

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
