# Peer review of "Blue Rubber Bleb Nevus Syndrome (BRBNS): A Rare Cause of Refractory Anemia in Children"

_children, 2022, doi:10.3390/children10010003_

Round 1

Reviewer 1 Report

This is an interesting report on a boy with BRBNS.

I think it is important to spread information on this syndrome since it is nowadays possible to diagnose it and treat it.

Would only suggest to improve English language in the introduction section. The rest of text is quite fine and well written.

Author Response

Dear reviewer :

I've used a paid editing service to revised my manuscript and wound soon submitted the manuscript again. Thank you for your kind suggestion. 

Reviewer 2 Report

The paper is thoroughly written and well researched.  It is a good case report.  There is a good discussion of the etiology, diagnosis and management of BRBNS.  

The timeline of how many times the patient came in and then left and lost to follow up is a bit unclear though.  That should be stated more clearly for the readers.  

However, the major problem with the paper is the English grammar is not correct in many of the sentences.  There probably over 100 sentences that need to be corrected.

Author Response

Dear reviewer :

(1) The patient came to our outpatient department for help 3 years ago and since then he would lost follow-up if no clinical symptoms or discomfort noted. He and his family finally agreed to undergo through examinations after recurrent severe anemia that required blood transfusion for multiple times about one and half year ago. After he was finally diagnosed as BRBNS, he had regular follow-up with oral iron supplement for 6 months. But he soon lost follow-up due to COVID-19 pandemic. We could only contact him and his family via phone.

(2) I've used a paid editing service to revised my manuscript and wound soon submitted the manuscript again. Thank you for your kind suggestion.